# Dissection of the *Fgf8* regulatory landscape by in vivo CRISPR-editing reveals extensive intra- and inter-enhancer redundancy

A. Hörnblad [1,2,3,4], S. Bastide[1,2,3,5,7,8], K. Langenfeld[1,8], F. Langa [6] & F. Spitz [1,2,3,7 ✉]

Developmental genes are often regulated by multiple elements with overlapping activity. Yet, in most cases, the relative function of those elements and their contribution to endogenous gene expression remain poorly characterized. An example of this phenomenon is that distinct sets of enhancers have been proposed to direct *Fgf8* in the limb apical ectodermal ridge and the midbrain-hindbrain boundary. Using in vivo CRISPR/Cas9 genome engineering, we functionally dissect this complex regulatory ensemble and demonstrate two distinct regulatory logics. In the apical ectodermal ridge, the control of *Fgf8* expression appears distributed between different enhancers. In contrast, we find that in the midbrain-hindbrain boundary, one of the three active enhancers is essential while the other two are dispensable. We further dissect the essential midbrain-hindbrain boundary enhancer to reveal that it is also composed by a mixture of essential and dispensable modules. Cross-species transgenic analysis of this enhancer suggests that its composition may have changed in the vertebrate lineage.

[1] Developmental Biology Unit, EMBL, Meyerhofstrasse 1, Heidelberg 69117, Germany. [2] (Epi)genomics of Animal Development Unit, Department of Developmental and Stem Cell Biology, Institut Pasteur, 75015 Paris, France. [3] UMR3738, CNRS, Paris, France. [4] Umeå Centre for Molecular Medicine, Umeå University, 90187 Umeå, Sweden. [5] Programme Doctoral Complexité du Vivant, Paris Sorbonne Université, Paris, France. [6] Mouse Genetics Engineering, Center for Innovation & Technological Research, Institut Pasteur, 75015 Paris, France. [7] Present address: Department of Human Genetics, The University of Chicago, Chicago, IL, USA. [8] These authors contributed equally: S. Bastide, K. Langenfeld. ✉email: fspitz@uchicago.edu

A fundamental feature of animal development is the dynamic and highly reproducible spatiotemporal expression of the genes that control cell fate. This spatial and temporal specificity is coordinated through the actions of *cis*-regulatory elements that can reside very far (up to Mb) from their target genes and even be located within neighbouring genes[1–6]. Transgenic studies have been important to identify enhancer sequences with regulatory activity in the genome[7], but this approach has a low throughput. More recently, next-generation sequencing approaches, such as chromosome conformation capture, ChIP-seq, DNAse-seq and ATAC-seq allowed for more comprehensive identification of candidate regulatory regions[1,2,4,8]. These studies have demonstrated that the regulatory architecture of developmental genes is complex: it frequently includes multiple regulatory elements, dispersed over large genomic regions that often display overlapping and/or redundant activity[9]. As useful they are, a strong limitation of these approaches is that they do not determine how important those elements are for gene expression. Indeed, it happens frequently that enhancers with strong transgenic activities have a surprisingly minor function in vivo in the control of their endogenous gene[10–13]. Because of this difference between function and activity, there is an urgent need to develop strategies to characterize the biological function of non-coding regulatory elements in vivo and in situ. Traditional gene targeting approaches have demonstrated the functional importance of individual enhancers, but the throughput of these techniques is relatively low[14–16]. Here, we deployed a Crispr/Cas in vivo genome-engineering approach to systematically dissect the functional importance of individual enhancers as well as their intrinsic logic in vivo, using the *Fgf8* locus as a model system.

FGF8 is a secreted signalling molecule with a highly dynamic gene expression pattern during development. It is essential for the normal development of the brain, craniofacial skeleton, limbs, and various other organs[17–22]. FGF8 is the key molecule for the formation and activity of the isthmic organizer (IsO) located at the midbrain-hindbrain boundary (MHB) between the mesencephalon and metencephalon[23–25] and that plays essential roles for patterning the midbrain and cerebellum[17,26]. Targeted deletion of *Fgf8* in the IsO leads to downregulation of MHB markers and subsequent loss of the midbrain and anterior hindbrain[26]. In the limb, *Fgf8* is expressed in apical ectodermal ridge (AER), at the distal tip of the limb bud. Absence of *Fgf8* leads to aberrant proximo-distal and anterior-posterior patterning, increased apoptosis in the limb bud and subsequent loss or hypoplasia of specific skeletal elements[20,21].

Although the consequences of *Fgf8* downregulation in the MHB and AER have been well characterized[20,21,26–28], less is known about the regulatory elements directing *Fgf8* expression in these structures. In a previous study, we characterized a 200 kb region forming the *Fgf8* regulatory landscape and identified three enhancers with the potential to drive expression in the mouse MHB and five enhancers that could drive expression in the limb AER[6]. The MHB enhancers are highly conserved from fish to mammals and two of them have indeed been identified as potential drivers of Fgf8 expression also in the zebrafish MHB[29]. The limb enhancers show a more diverse degree of conservation but all of them are conserved at least from amniotes to mammals[6].

In this study, we address the in vivo contribution of these two sets of enhancers to *Fgf8* expression in the limb and the MHB, respectively. Using in vivo CRISPR/Cas9 genome editing, we demonstrate extensive redundancy between enhancers in the limb, while in the MHB, one distant primary enhancer is essential for *Fgf8* expression. We further dissect the main MHB enhancer extensively to identify its functional units and define two essential

subunits required for its function. Intriguingly, although deletion of only 37 bp is enough to abrogate the regulatory potential of this enhancer and cause loss of midbrain and cerebellar structures, we also reveal widespread functional redundancy within this essential enhancer. Furthermore, we demonstrate that albeit sequence conservation may predict similar enhancer activity in fish and mouse, the functional subunits of the enhancer appear to have diverged and reorganized their regulatory logic.

## Results

**Extensive regulatory redundancy for *Fgf8* expression in the limb.** A previous study identified a set of putative limb and MHB enhancers in the *Fgf8* locus with the potential to drive gene expression in these tissues[6] (Fig. 1a). In order to investigate their role in vivo, we generated mice with targeted deletions of each individual enhancer as well as compound deletions of the two proximal MHB enhancers. To this end, we performed zygote injections of Cas9 mRNA and two chimeric gRNAs flanking the regions of interest (Supplementary Fig. 1, Supplementary Table 1 and methods, in vivo deletion efficiency ranging from 4 to 40% in born pups). We assessed the consequence of these enhancer deletions in hemizygous condition over *Fgf8* null alleles (either Fgf8[null/+] [17] or DEL(P-F8)[6]).

For the limb, four enhancers (CE58, CE59, CE61, CE66) are spread within a 40 kb region in the introns of the neighbouring *Fbxw4* gene while only CE80 is located in the proximity of *Fgf8* (Fig. 1a). Previous experiments had demonstrated that mice carrying a deletion of the region containing the four distal enhancers abolishes limb *Fgf8* expression and causes similar defects to the conditional ablation of *Fgf8* in the limb[6]. All the mutants that we generated carrying single deletions of these putative enhancers were healthy and fertile and did not display any apparent developmental phenotypes. Importantly, the limbs were indistinguishable from their control littermates. These results were confirmed in more detail by skeletal preparations of e18.5 embryos (Fig. 1c). We also analysed the expression pattern of *Fgf8* at e10.5 using in situ hybridisation. At this stage *Fgf8* is strongly expressed in the morphologically well-defined AER of both the forelimb and the hindlimb (Fig. 1b). The AER expression pattern displayed by embryos carrying enhancer deletions was indistinguishable from their control littermates (Fig. 1d).

To further confirm this, we performed quantitative RT-qPCR analysis on dissected e10.5 forelimbs of three deletion lines (DEL58, DEL61, DEL80, corresponding to deeply evolutionary conserved enhancers) and failed to detect significant change in *Fgf8* gene expression levels or in other limb patterning genes, which could have indicated compensatory effects (Supplementary Fig. 2). Thus, from a pure functional viewpoint, each of those enhancers appears dispensable for the expression of *Fgf8* and subsequent development of the limb. Taken together, this suggests that the regulatory system that controls *Fgf8* limb expression in vivo is highly modular and displays extensive regulatory redundancy.

**A distant *Fgf8* enhancer is required for formation of the midbrain and cerebellum.** In the MHB, two of the putative enhancers (CE79 and CE80) are located within a 20 kb region downstream of *Fgf8*, while the third one (CE64) is located at a distance of 120 kb within an intron of the neighbouring gene *Fbxw4* (Fig. 1a). Using CRISPR/Cas9 zygote injections, we generated mice carrying single deletions of these enhancers as well as the double deletion of CE79 and CE80. We found no morphological differences between the DEL79, DEL80 or the compound DEL79-80 animals and their control littermates that could be

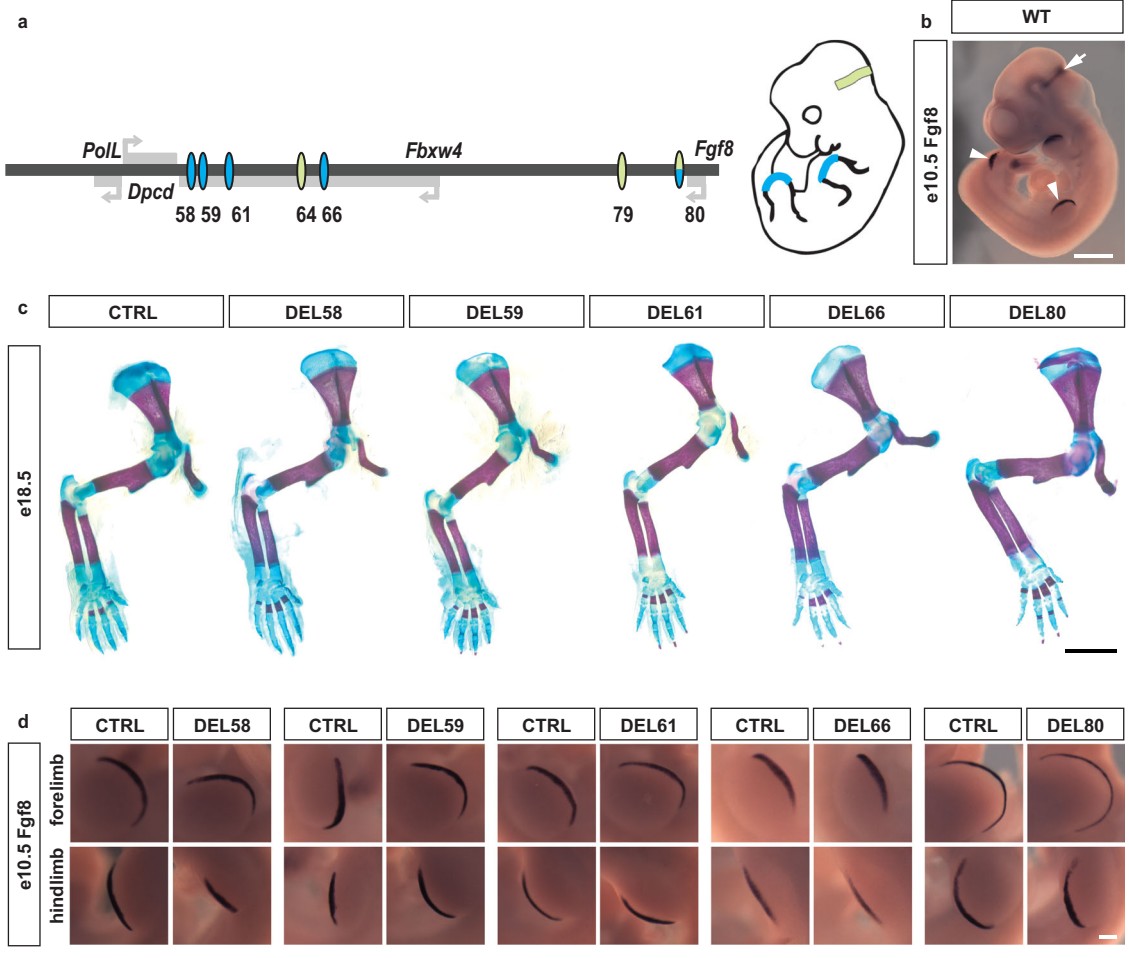

**Fig. 1 The limb AER presents extensive regulatory redundancy. a** Schematic representation of the two sets of conserved elements directing expression in the AER (blue), and in the MHB (green). **b** In situ hybridization with riboprobe against *Fgf8* mRNA (*n* > 3). Arrowheads and arrow indicate AER and MHB, respectively. **c** Photomicrograph of alizarin-red/alcian-blue stained e18.5 forelimbs from control and AER enhancer deletion embryos (*n* > 3). **d** In situ hybridization of control and AER enhancer deletion embryos at e10.5 with riboprobe against *Fgf8* (*n* > 3 for all experimental groups, with the exception of DEL66 where *n* = 3). All mutant embryos display expression patterns indistinguishable from their littermate controls. Scale bar is 1 mm in (**b**), 2 mm in (**c**), and 200 μm in (**d**).

detected macroscopically in the brain nor in other tissues and the mice were viable and fertile in homozygosis. In contrast, DEL64 mice display a complete absence of midbrain and cerebellar structures visible at e18.5, phenocopying the conditional KO of Fgf8 in the MHB[26]. A more detailed analysis of e18.5 brains using optical projection tomography (OPT) demonstrates the complete loss of superior colliculus, inferior colliculus, isthmus and cerebellum in the DEL64 mutants (Fig. 2, Supplementary Movie 1). These analyses also confirmed the normal appearance of these structures in the DEL79, DEL80, and DEL79-80 mutants (Fig. 2, Supplementary Movie 2–4). In summary, of the three MHB enhancers, only CE64 is essential for proper development of the MHB.

**Deletion of CE64 completely abolishes *Fgf8* expression in the MHB.** We further explored the spatial expression of *Fgf8* at e10.5 in all the generated MHB mutants (Fig. 3a–f). At this time point in development, the expression of *Fgf8* has been narrowed down to a sharply delimited band of cells at the border the midbrain and anterior hindbrain. In the DEL64 embryos *Fgf8* expression was completely absent in the MHB and the morphology of these embryos already revealed the absence of a large portion of the midbrain (Fig. 3b). In DEL79, DEL80 and DEL79-80 embryos,

*Fgf8* expression pattern and signal strength were similar to control embryos. Next, we performed in situ hybridisation analysis of *Fgf8* expression at the earliest stage of expression, e8.25, in DEL64 embryos. These analyses revealed a complete lack of *Fgf8* expression also in the initial expression phase (Fig. 3g, h). Together, these experiments demonstrate that CE64 is required and sufficient for proper initiation of *Fgf8* expression and sufficient for subsequent maintenance in the developing MHB. We also found that a subtle decrease in signal intensity also appeared to be present in the forebrain and the primitive streak, indicating that CE64 may play a role also in these tissues. Consistent with this possibility, CE64 showed in transgenic assays that it could drive strong LacZ reporter gene expression in the forebrain (in a domain much broader than the actual *Fgf8* expression domain. However, DEL64 embryos do no show the telencephalic defects observed in *Fgf8* mutants or deletion of its distal regulatory region (DEL(P4) {Marinic et al.}, indicating the predominant forebrain enhancer(s) is still distinct and largely independent from CE64.

Although in situ hybridisation revealed similar expression patterns between DEL79, DEL80 and DEL79-80 mutants as compared to control embryos, we sought to assess potential subtle quantitative changes in the expression levels, by RT-qPCR on dissected MHB region from e10.5 embryos.

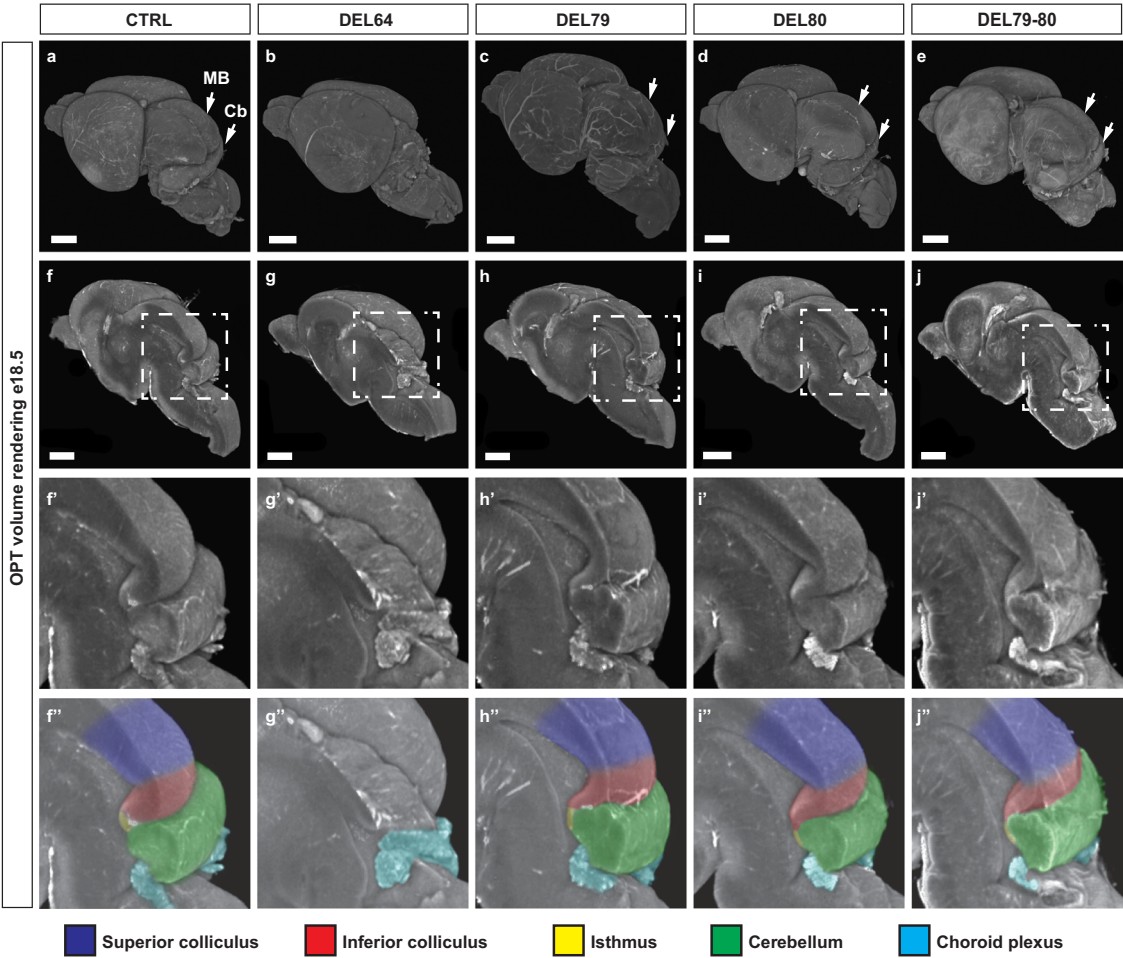

**Fig. 2 One main enhancer is required for Fgf8 expression in the MHB. a–j″** OPT generated volume renderings of e18.5 brains from control (**a**, **f**, **f′**, **f″**), DEL64 (**b**, **g**, **g′**, **g″**), DEL79 (**c**, **h**, **h′**, **h″**), DEL80 (**d**, **i**, **i′**, **i″**) and DEL79-80 (**e**, **j**, **j′**, **j″**) mutants. Signal is based on tissue autoflourescence. Control brains, DEL79, DEL80 and DEL79-80 mutants display a well-developed midbrain and cerebellar anlage while DEL64 brains display severe hypoplasia of midbrain and cerebellum. **f–j** Midsagittal digital dissection reveal complete loss of all MHB derived structures in the DEL64 mutant. **f′–j″** Close-up of boxed area in (**f–j**). Brains have been pseudocolored in (**f″–j″**): dark blue, superior colliculus; red, inferior colliculus; yellow, isthmus; green, cerebellum; light blue, choroid plexus. Scale bar in (**a–j**) is 1 mm. MB, midbrain; Cb, cerebellum.

In the controls for these experiments, we noticed that mice heterozygous for a null *Fgf8* allele did not show halved expression levels as expected but a much milder reduction of *Fgf8*: in the MHB of $Fgf8^{null/+}$, *Fgf8* expression was 79% of wild-type level (Fig. 3i, Supplementary Fig. 2). This limited impact suggests that compensatory mechanisms may up-regulate *Fgf8* mRNA levels in response to a decrease in gene dosage. In $Fgf8^{null/+}$ MHB, we found a decreased expression of *Spry2* and *Dusp6* (Supplementary Fig. 3), two downstream targets of *Fgf8* that have been suggested to be part of negative feedback loops for Fgf-signalling[30,31]. We speculate that activation of this feedback circuit could account for sustained expression upon *Fgf8* gene dosage reduction at e10.5 in the MHB and could possibly shadow direct effects of enhancer deletion. Hence, we included in our expression analysis of the mutants, several genes part of the *Fgf8* pathway, with the idea that the impact of *Fgf8* expression perturbation may be better reflected by alterations in the network state than the single expression of *Fgf8* itself

In the DEL79 embryos, *Fgf8* expression level appear unaffected (Fig. 3i), but they display minor up-regulation of genes (most significantly *Fgf17*, *Pax2*) which may indicate activation of a compensatory system (Supplementary Figs. 3–S4). In DEL80 as well as the compound DEL79-80, we could detect a mild but significant decrease in expression of *Fgf8* as compared to the

control animals (Fig. 3i). This decrease was accompanied by a small but significant downregulation of other genes in the MHB regulatory network (Supplementary Figs. 3–S4). Taken together, CE64 appears as the main enhancer of *Fgf8* expression in the MHB. Despite the sensitivity of the MHB-derived structures to mild-reduction of Fgf8-signalling from the IsO[17,28], we did not see any phenotypic defects in absence of CE79 and CE80, indicating that their input appear mostly dispensable for MHB patterning and development. However, we see indication suggesting small direct effects resulting from the deletion of CE79 and CE80 that may be compensated by overall re-calibration of a gene regulatory network canalizing phenotypic variations due to mild expression changes.

**Temporal specificity of CE64 underlies initiation of *Fgf8* gene expression.** The absence of a detectable function of CE79 and CE80 is surprising, given their high conservation in vertebrates. In zebrafish, the ortholog of CE79 does not demonstrate enhancer activity during the earliest timepoints of *Fgf8* expression in the MHB[32]. In mice, our experiments show that CE64 is necessary for the early activation of *Fgf8*. We hypothesize that one possible explanation to the singular functional role of CE64 amongst the three MHB enhancers can be related to specific differences in the

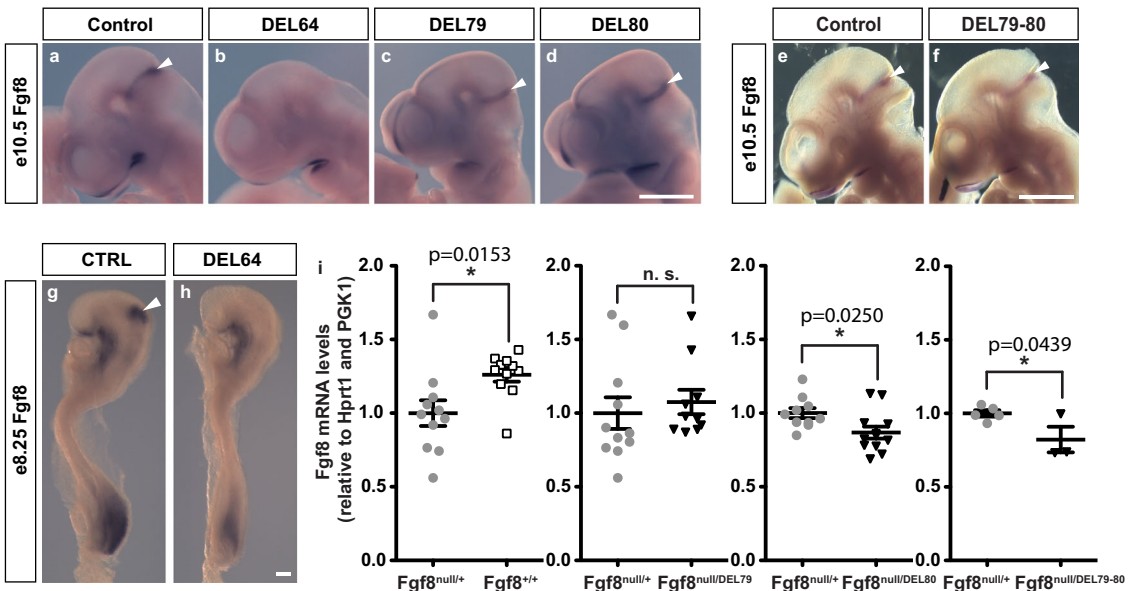

**Fig. 3 CE64 is required and sufficient for initiation and maintenance of *Fgf8* expression in the MHB. a–f** In situ hybridization against *Fgf8* mRNA in e10.5 embryos from control (**a, e**), DEL64 (**b**), DEL79 (**c**), DEL80 (**d**) and DEL79-80 (**f**) mutants (*n* > 3 for all experimental groups). Note the complete lack of Fgf8 expression in the MHB of DEL64 (**b**) embryos as compared to controls (**a**). Brain tissue is markedly reduced already at this stage in DEL64 embryos (**b**). **g, h** In situ hybridization of control and DEL64 enhancer deletion embryos at e8.25 with riboprobe against *Fgf8*. No expression is detected in the DEL64 embryos (*n* = 4). **i** Relative expression of *Fgf8* mRNA levels in dissected MHB region from WT (*n* = 11), DEL79 (*n* = 10), DEL80 (*n* = 12), and DEL79-80 (*n* = 3) e10.5 embryos as compared to Fgf8$^{null/+}$ (*n* = 11, 11, 10, 5) control littermates. Individual data points, mean ± SEM are indicated. *$p < 0.05$, n.s. = not significant (two-tailed Student's *t*-test; exact *p*-values are indicated in the figure panel). Scale bar is 1 mm in (**a–f**) and 100 μm in (**g, h**).

timing of their activation during the initiation phase of *Fgf8* expression, while they may later be redundant. To test this and limit the bias of position effects, we performed transient transgenesis of reporter constructs driven by CE64, CE79 or CE80 and analysed series of embryos at early stages of development. For transgenes driven by CE64, we could detect reporter expression in the MHB as early as the 5-somites stage while we only saw MHB reporter expression for CE79 and CE80-driven transgenes at later stages (15 and 13 somites respectively, Supplementary Fig. 5). Even though this observation should be taken with caution given the low number of embryos and the sensitivity of the reporter to position effects (which can impact timing of reporter activation in an embryo-specific manner), it is consistent with a later activation of CE79 and CE80. This timing difference may lead to a critical window of time during which CE64 is the only MHB active enhancer, which could account for its prominent functional role in activating *Fgf8*.

**In vivo CRISPR/*Cas9* screen identifies two distinct subunits required for CE64 enhancer function.** Given the crucial role of CE64 for the expression of *Fgf8* in the MHB we aimed to dissect how the regulatory logic of this enhancer is composed in vivo. To this end, we injected a new set of CRISPR gRNAs in different combinations together with *Cas9* mRNA in oocytes that had been in vitro fertilized using sperm from males heterozygous for the DEL(P-F8) allele (Fig. 4a). With this approach, half of the injected embryos are hemizygous for the *Fgf8* locus, which greatly increase the yield of informative embryos and their identification, and reduce the potential cofounding case of mosaicism. It also allowed us to directly screen F0 embryos at e18.5 for midbrain or cerebellum hypoplasia (Fig. 4b) and identify regions contributing to CE64 function. Using this strategy, we produced and analysed a large collection of deletions spanning different regions of CE64 at the endogenous locus. This allowed us to perform in vivo and in situ "enhancer-bashing" experiments, focusing on enhancer

function in its endogenous context (and not measuring its out of context activity). All embryos produced were genotyped by PCR for targeted deletions and the breakpoints were sequenced. In addition, the embryos carrying deletions were genotyped with primers internal to the identified deletions in order to discard embryos carrying WT alleles due to mosaicism (Fig. 4c). In all, we identified 39 informative alleles (Supplementary Table 2).

This extensive panel of deletions allowed us to define three distinct elements in CE64, of which one is dispensable (64-A in Fig. 4e) and two (64-B and 64-C in Fig. 4e) are essential and required for proper enhancer function. Deleting any of the two essential regions 64-B or 64-C is sufficient to completely abrogate the development of the midbrain and anterior hindbrain region. Of these essential subunits, 64-B spans a region of ~700 bp that is highly conserved among vertebrates (Fig. 4d). Deletions of subregions in 64-B demonstrated that considerable functional redundancy exists within this subunit. In fact, deleting two-thirds of 64-B is not sufficient to abolish proper midbrain and cerebellum formation (DEL-B3 in Fig. 4d, region R2-R3 in Fig. 4e) and any one third of this subunit is dispensable for its function (DEL-B2, DEL-B4, DEL-B5 in Fig. 4d corresponding to R1, R2 and R3 in Fig. 4e). Therefore, it seems that the regulatory information embedded in 64-B is modular and spread across the element, rather than organised as one continuous regulatory unit. Subunit 64-C is only 180 bp long and located on the most telomeric side of CE64. It is conserved in tetrapods but not in fish. Consecutive deletions of sub-regions in 64-C do not cause any phenotype (DEL-C2, DEL-C3, DEL-C4 in Fig. 4), but remarkably, the deletion of merely 37 bp in 64-C at the junction between 64-C2 and 64-C3 is sufficient to completely abrogate CE64 function (DEL-C5 in Fig. 4a). This indicates that the 37 bp contains at least two critical, yet redundant elements.

**The functional subunits of CE64 are interdependent.** Next, we asked if 64-B and 64-C differ in their regulatory potential by

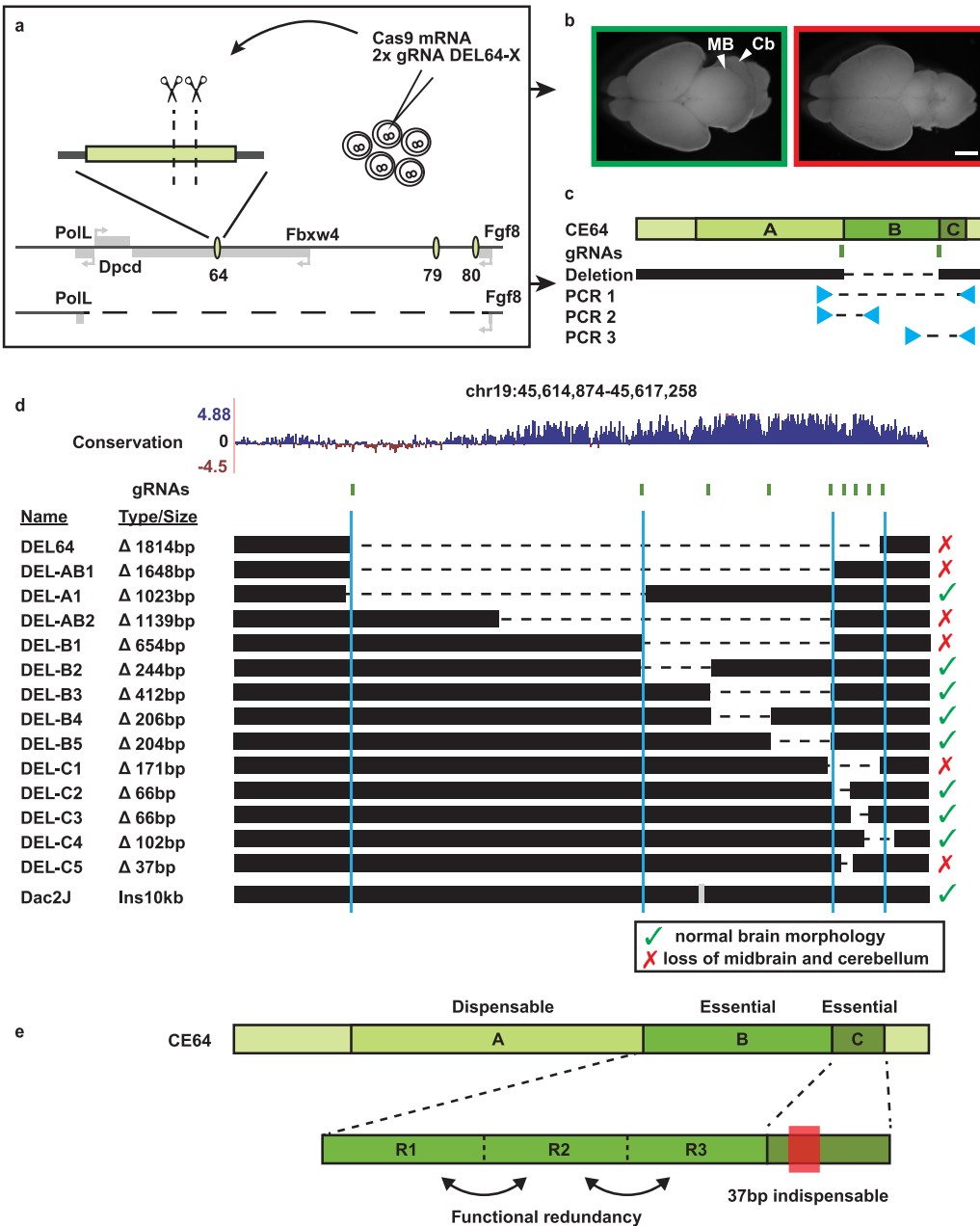

**Fig. 4 Two distinct subunits with internal redundancy are required for CE64 function. a** Schematic representation of the CRISPR screen setup. Oocytes were fertilized with males carrying one DEL(P-F8) allele (bottom) and 2 gRNAs were simultaneously injected with *Cas9* mRNA. **b** Brain morphology of F0 embryos was examined at e18.5 and all embryos were genotyped according to strategy in **c** to identify breakpoints and possible mosaicism. **d** Representation of the panel of deletions generated. The in vivo CRISPR/*Cas9* screen defined two indispensable subunits of CE64 (DEL-B and DEL-C). Removing overlapping bits within these units (DEL-B2 through DEL-B5, DEL-C2 through DEL-C4) does not provoke any phenotype. The smallest deletion causing lack of MHB derived structures is merely 37 bp (DEL-C5). Red cross indicates loss of MHB derived tissues and green tick indicates normal brain morphology. Vertical grey dash indicates location of 10 kb insertion in Dac2J mice. **e** Schematic representation of the functional units of CE64. Both 64-B and 64-C are indispensable for CE64 function, while 64-A is not required. Functional redundancy is encoded within these subunits although a deletion of only 37 bp is enough to abrogate the function of 64-C.

performing transient transgenesis of a reporter construct carrying either CE64, 64-B or CE64 lacking 64-B or 64-C, respectively (Fig. 5). As expected, Tg(CE64) recapitulated the expression pattern published for CE64 in 3 out of 4 transgenic embryos (Fig. 5c and Supplementary Fig. 6, for comparison see Marinic et al.[6]). However, for both Tg(DEL-B) and the Tg(DEL-C), no expression was detected in the MHB (0/8 and 0/3 embryos respectively) (Fig. 5c, Supplementary Fig. 6). Some of the Tg(DEL-B) embryos (4/8) displayed a reproducible reporter expression in the anterior

hindbrain (Supplementary Fig. 6). This may indicate that 64-C has an intrinsic regulatory potential that is independent of 64-B for expression per se but which spatial position is shifted in presence of 64-B. In contrast, 64-B does not appear to have any autonomous activity in e10.5 embryos (0/4 embryos) (Fig. 5c, Supplementary Fig. 6). Taken together, the transgenic assays indicate that although both 64-B and 64-C are required for the function of CE64, their intrinsic properties are not sufficient to drive spatial expression in the MHB on their own.

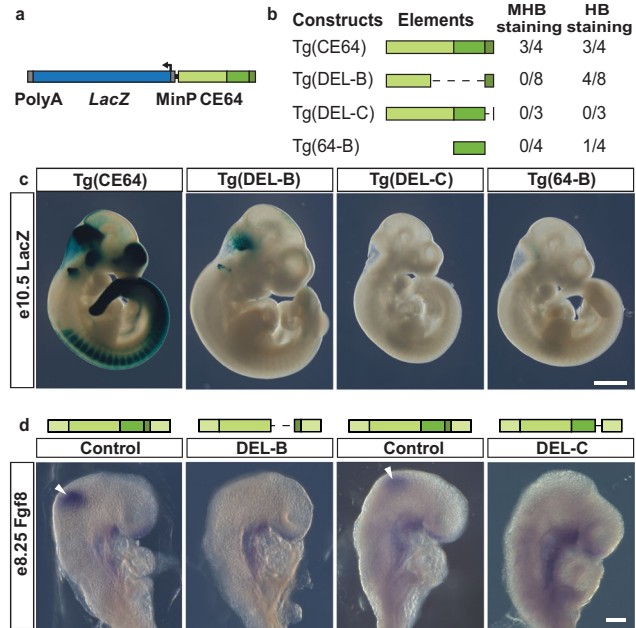

**Fig. 5 The function of 64-B and 64-C is interdependent. a** Schematic of transgenic reporter construct containing *LacZ*, a minimal promoter and the putative enhancer sequence of interest. **b** Table of constructs used for transgenesis, the CE64 subunits included and the number of embryos displaying reporter expression for each construct (i.e. 3/4 means that out of a total of 4 transgenic embryos, 3 display positive reporter expression). **c** Photomicrographs of representative embryos stained for *LacZ* activity. Reporter expression in the MHB is only detected in wt Tg(CE64) embryos. Note the staining that is present in the anterior hindbrain of some of the Tg (DEL-B) embryos. Tg(DEL-C) and Tg(64-B) do not manifest any reproducible reporter expression. **d** In situ hybridization of control, DEL-B and DEL-C deletion embryos at e8.25 with riboprobe against *Fgf8*. Expression is undetectable in both enhancer subunit deletions (*n* = 4). Blue box indicates *LacZ* reporter gene in (**a**). In all panels, light green, green and dark green boxes indicate 64-A, 64-B, and 64-C subunits, respectively. Scale bar is 1 mm in (**c**) and 100 μm in (**d**).

The transient transgenesis and CRISPR/*Cas9* deletion screen of F0 progeny at late stages of embryonic development precluded direct analysis of *Fgf8* gene expression at the time when the MHB patterns the prospective midbrain and hindbrain. In order to assess the contribution of each of the CE64 sub-units to *Fgf8* expression in the MHB during early stages we generated mouse lines carrying deletions of 64-B and 64-C. As expected, both lines completely lack midbrain and cerebellum at e18.5, confirming the results from the embryonic screen. Expression analysis by in situ hybridisation at e8.25 demonstrated that in both mutants, *Fgf8* gene expression fails to initiate and is completely absent from the MHB region (Fig. 5d). Altogether, these data demonstrate that the functional elements of CE64 are units that have reciprocal dependency in order to mediate proper regulatory input to *Fgf8*.

**Evolutionary conservation of CE64 sequence versus functional organisation.** Conservation is a good predictor for identifying regulatory regions in the genome and a previous study has shown that the zebrafish region orthologous to CE64 can drive expression in the zebrafish MHB (dr10 in ref. [29]). Intriguingly, our functional analysis in mouse of CE64 sub-regions identified an essential part of the enhancer (64-C) that is not conserved in fish (Fig. 6a). In addition, transgenic analysis showed that the conserved 64-B element is unable to drive expression in the MHB by its own (Fig. 5c). We, therefore, asked whether the orthologous

region in fish could drive MHB expression in the mouse. To this end, we cloned CE64 from spotted gar, a species that is closer to mouse and humans in the vertebrate lineage and has not undergone the genome duplication that the teleost lineage has. Remarkably, the 350 bp sequence from spotted gar could drive expression in the MHB region in 4 out of 4 embryos (Fig. 6b, Supplementary Fig. 6), despite lacking a region orthologous to the mouse 64-C. The expression did not completely reproduce the expression of the full CE64 but was restricted to the MHB and dorsal part of the anterior hindbrain. It is also noteworthy that the zebrafish dr10 enhancer recapitulates the broad activity of mouse CE64 in the MHB region (as well as in the forebrain and tail bud), in the zebrafish transcriptional context[29]. This raises the question to whether non-conserved sequences outside the 350 bp core enhancer may encode additional information that would further increase the similarity in regulatory potential to mouse CE64.

To investigate the sequence composition of CE64, we used multiple alignments to define phylogenetic footprints, hence identifying highly conserved sub-regions of the enhancer that might represent where functional TF binding can occur. In 64-B the high conservation of the sequence precluded identification of obvious putative TFBS, while in 64-C we could define 4 conserved blocks of sequences resembling TFBS or TFBS clusters in length and composition (Fig. 6c, blue boxes, see alignments in Supplementary Fig. 7). The 37 bp deletion in 64-C abrogates two of these conserved blocks (red box in Fig. 6c and d), demonstrating that they are functionally important. Block #2 shares similarities with TCF/LEF binding sites (Fig. 6c), which can mediate responsiveness to Wnt-signalling, a known upstream inducer of *Fgf8* expression in the MHB[33,34]. Noteworthy, mouse 64-B also comprises a potential Wnt-TCF/LEF response element (sequence CAGTTTCAAAGGAA). Block #3 bears homologies to the consensus binding motif defined for En1/2 (Fig. 6c), two transcription factors specifically expressed in the MHB[35,36] and that contribute to *Fgf8* maintenance there[37], as well as to some extent to Sox proteins (Fig. 6c).

We then used these footprints to derive positional weight matrices (PWMs) and scan the spotted gar and zebrafish CE64 for corresponding motif occurrences. Only one of the two PWMs derived from the phylogenetic footprints (block #2 and #3, Fig. 6c) in the 37 bp deletion was detected in the spotted gar (Supplementary Fig. 8A) or the zebrafish (Supplementary Fig. 8B) CE64 (including the whole sequence tested in ref. [29], and its spotted gar ortholog). These analyses open up for the possibility that orthologous CE64 elements that drive MHB expression in the teleost fishes and mammals could use different logics that may correspond to a rewiring of the *Fgf8* regulatory circuit.

## Discussion

Shadow and distributed enhancers have been described as common features in the regulatory genome that could provide robustness to gene expression, by buffering it against environmental changes and possible genetic variation[38–40]. The *Fgf8* regulatory landscape is a prototypical example of the complexity of developmental gene regulation, which involves multiple enhancers with similar activity. By dissecting their function in vivo we found different acting logics within two sets of tissue-specific enhancers. In the limb, *Fgf8* AER expression results from the collective action of several enhancer modules with redundant activity (Fig. 7a). Similarly to a recent study of other limb enhancers[40], we fail to detect gene expression changes for single *Fgf8* enhancer deletions in the limb. This contrasts somewhat with few studies on redundant enhancers in other mouse tissues where differential quantitative changes can be detected upon

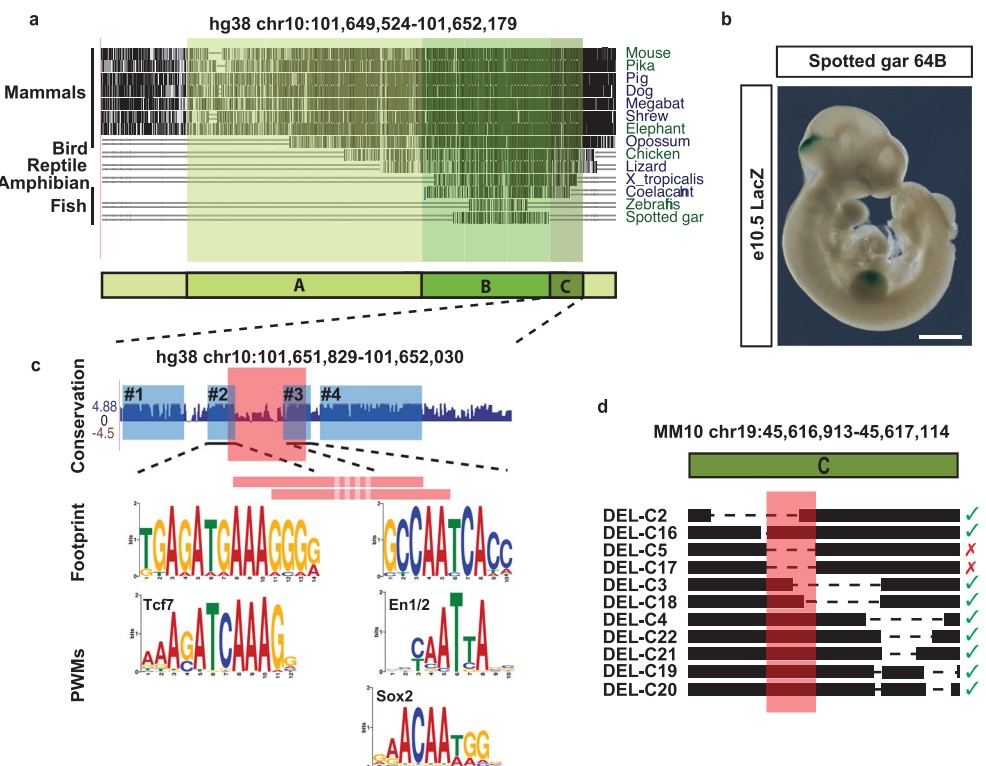

**Fig. 6 Cross-species comparison reveals non-conserved essential features of mouse of CE64. a** Sequence conservation of CE64 across species. 64-B is conserved from fish to mammals while 64-C is conserved among tetrapods. **b** Photomicrograph of a transgenic embryo injected with a minimal reporter construct including spotted gar CE64 and stained for *LacZ* activity (*n* = 4, of which 4 displayed positive expression). Note that only 64-B is conserved in the spotted gar CE64. Arrowhead indicates the MHB. **c** Upper panel: sequence conservation score of 64-C. Blue boxes indicate highly conserved blocks. Red box indicate the smallest deletion that abrogates CE64 function. Middle panel: phylogenetic footprints generated from multiple sequence alignments corresponding to conserved block #2 and #3. The red bars indicate the breakpoints of the two smallest phenotype-causing deletions. Lower panel: PWMs of Tcf/Lef1, En1/2 and Sox proteins display similarities to the generated phylogenetic footprints. **d** Overview of small deletions in 64-C from the CRISPR/Cas9 screen. Red box indicates 37 bp depicted in (**c**). Red cross indicates loss of MHB derived tissues and green tick indicates normal brain morphology. Scale bar in **b** is 1mm.

deletion of single enhancers[41,42]. Based on our data, we cannot attribute a higher degree of functional importance to any of the different AER enhancers, although compound deletion of the four distal enhancers (excluding CE80) abolishes *Fgf8* expression in the AER[6]. Still, this does not exclude that the individual *Fgf8* AER enhancers may function additively and play slightly different roles for gene expression. One possibility would be that there is a minimum threshold for the frequency of enhancer-promoter interactions that is required for normal transcription of *Fgf8*, and that despite removing one enhancer this frequency is maintained at sufficiently high levels. Noteworthy, we do not see evidence that the modules conserved only in tetrapods (CE61, CE66) contribute specifically to the heterochronic shift associated with evolution of the AER from a primitive apical ectodermal fold[43]. Contrarily to a simple view, the progressive recruitment of new AER enhancer modules during tetrapod evolution did not simply reinforce expression by addition of accessory elements to an ancestral essential enhancer. Enhancer multiplicity may have allowed a redistribution of functional roles between the new elements, enabling more complex rewiring of the expression control of this gene in the apical ectoderm of the limb, which could have contributed to a prolonged maintenance of the apical ectodermal ridge, an essential step in the evolution of tetrapod limbs[43,44].

In the MHB, early *Fgf8* expression is dependent on one enhancer and the others appear dispensable (Fig. 7a). CE64 appears to be the first active MHB enhancer, and hence is critical to initiate *Fgf8* expression in this structure, ensuring cell survival in the mesencephalon/metencephalon region[26] and maintenance of expression of critical mes/met transcription factors, such as EN1/2, PAX2/5[26], which could, in turn, regulate *Fgf8* expression, possibly through different enhancers such as CE79[32]. Given the high conservation of CE79 and CE80 and their previous identification also as putative enhancer in the MHB in the zebrafish[29,32,45], the finding that both are dispensable for normal development of the MHB region may be surprising. Still, it remains to be defined if those enhancers have important roles in other embryonic structures or later stages, and whether they may contribute to aspects of MHB development in the context of genetic or environmental challenges. MHB development is particularly sensitive to *Fgf8* dosage and expression length[27] and is known to be sensitive to genetic context. For example, it has been shown that deletion of the MHB key TF *Pax2* in mice leads to complete loss of *Fgf8* MHB expression and associated anatomical structures on the C3H/He genetic background while mutants in the C57Bl/6 background do not display these phenotypes[46,47], indicating the role of genetic variation in buffering potential defects associated with *Pax2* knockout, in this case through possible expression timing of other redundant Pax TF family members. Hence, it is possible that CE79 and CE80 may play significant roles in MHB development by maintaining *Fgf8* expression levels and sustaining its expression for longer periods of time, albeit in different genetic or environmental contexts than the ones tested here.

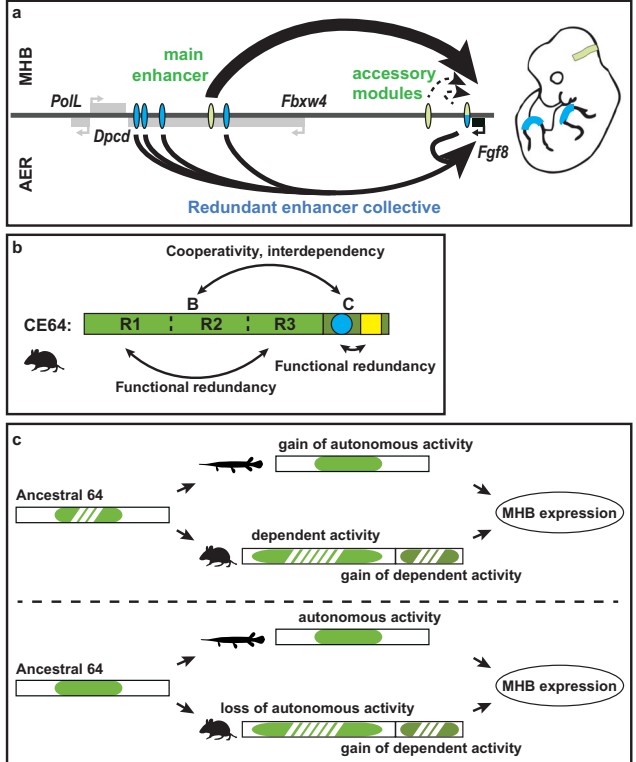

**Fig. 7 Two modes of regulatory redundancy provide robustness to *Fgf8* gene expression. a** Schematic representation of enhancer activity in the MHB and the AER. In the MHB, one main enhancer is required and sufficient to direct *Fgf8* expression (upper panel). In the limb AER, a collective of redundant enhancers, each by themselves dispensable, directs gene expression (lower panel). Blue ovals: AER enhancers, green ovals: MHB enhancers. **b** Schematic of the mouse CE64 and its subunits. CE64 is composed of two essential regulatory units that are reciprocally dependent and cannot alone direct expression in the MHB. Both of the regulatory subunits exert functional redundancy within themselves. This redundancy may be encoded by similar TFBS or recurrent DNA motifs in 64-B, while in 64-C it is encoded by two distinct DNA signatures (blue circle, yellow square) that reciprocally can buffer the loss of each other. R1 to R3 indicates redundant blocks within 64B. **c** Two scenarios for CE64 evolution. Upper panel: both spotted gar and the tetrapod lineage independently gained MHB regulatory activity, spotted gar within the ancestral enhancer (64-B in mouse) and tetrapods by addition of a new subunit (64-C). Lower panel: spotted gar CE64 retain an autonomous MHB regulatory activity in the absence of 64-C while in the tetrapod lineage, 64-B appears to have lost its autonomous activity and 64-C has been added to the regulatory wiring. Green area represents autonomous regulatory activity and dashed green non-autonomous activity.

The location of CE64 in an intron of *Fbxw4*, a gene flanking *Fgf8*, raises the question whether the MHB phenotype associated with CE64 may in part result from an impact of CE64 deletion on the function of *Fbxw4*. The complete absence of the MHB in DEL64 mutants precludes the analysis of quantitative gene expression changes of *Fbxw4* in these cells. Yet, genetic evidence argues against a significant involvement of *Fbxw4* in the observed phenotypes. Firstly, mice heterozygous for DEL64 over a Fgf8 null allele (ie. where the *Fbxw4* locus is intact) showed the same MHB aplasia than homozygous DEL64 mutants. In contrast, other intronic deletions within *Fbxw4* (CE58, CE59, CE61 and CE66, the latter two in the same intron as CE64) had no detectable impact on MHB development, either in homozygous conditions or over a *Fgf8* null allele. Furthermore, a 10 kb insertion of a

mouse ERV just next to CE64 enhancer which led to a premature termination of *Fbxw4* transcripts[48,49] do not interfere with MHB morphogenesis. All these observations argue that the main effect of deleting CE64 is intrinsic to its function as an enhancer for *Fgf8* and not due to a by-standing effect on *Fbxw4*.

Thorough dissection of the functional units of CE64 reveals a multi-layered organization, with separate units critical for its activity (Fig. 7b). The failure to initiate expression when deleting either of these regions demonstrates that their activities are interdependent (Fig. 7b). Our extensive in vivo screen of smaller deletions within CE64 nonetheless suggests that this enhancer can withstand relatively large sequence modifications, even in its evolutionary conserved parts. The small 37 bp region, which deletion completely abrogates the function of the main MHB enhancer, thus causing loss of midbrain and cerebellum, identifies an essential and compact part of this enhancer. As removing overlapping bits of these 37 bp does not lead to any phenotype, it demonstrates that functional redundancy is encoded in the regulatory architecture of the enhancer, involving most likely two sets of factors (Fig. 7b). Sequence analysis suggests that Wnt-mediators LEF/TCF and EN1/2 or SOX may be the transcription factors associated with this activity.

The comparison of CE64 elements from different species showed that one of the two critical regions we identified is only present in tetrapods, which suggests that CE64 may use different logic in different lineages. The interdependence between 64-B and 64-C in the murine enhancer may have been acquired late during tetrapod evolution and may correspond to a change in *Fgf8* regulation. The fact that CE64 from spotted gar, in contrast to mouse 64-B, can drive expression autonomously in the MHB boundary in transgenic mice suggests that the spotted gar subunit either gained new regulatory potential or that an ancestral regulatory potential has been lost in the mouse enhancer subunit (Fig. 7c). Evolution of regulatory potential adjacent to the ancestral CE64, which led to the 64-C sub-enhancer element in the tetrapod lineage may have allowed for loss of autonomous regulatory potential in the ancestral 64-B, (Fig. 7c). It would be interesting to see if these changes were strictly compensatory (using the same TFs lexicon but just rearranged on different sites) or may correspond to a rewiring of *Fgf8* regulation (by recruiting new TFs). Altogether, the dissection of CE64 shows that it follows a complex logic involving multiple modules, which can both contribute to set up the very specific expression pattern of *Fgf8* in a given species in a robust manner, but as well allow for flexibility and functional changes on evolutionary timescales.

The complexity of developmental regulatory ensembles and enhancer elements has always made their functional studies difficult. Here we demonstrate that Crispr/Cas9 in vivo deletion-screens can be very efficient in functionally dissecting their constituents. If several high-throughput screens have been conducted in cell lines using CRISPR/*Cas9*[50–56] and some studies have addressed enhancer redundancy by enhancer deletion in mouse[40–42,57], our study shows that systematic dissection of enhancer function can be carried out in vivo in mouse embryos. The use of a large deletion to allow hemizygous conditions is not mandatory, but provide both increased yield and easier analysis. By focussing on function in situ and not on activity out of context, our approach provides an important complement to the transgenic enhancer bashing assays that has been performed so far. Such an approach is particularly necessary, given the intricate interplay between different units or enhancer modules, both at large scale within an ensemble and within an enhancer and may reveal features or activities embedded in enhancers such as their range of activity[1,58], cooperativity[59] and additivity[60], which cannot be assessed otherwise. Our study illustrates the feasibility and usefulness of such approaches to decipher the

complex, flexible and multi-scale organisation of developmental gene regulatory ensembles.

## Methods

**Animals and genotyping**. All animal procedures were performed according to principles and guidelines at the EMBL Heidelberg (Germany) and the Institut Pasteur (Paris, France), as defined and overseen by their Institutional Animal Care and Use Committees, in accordance with the European Directive 2010/63/EU. Genotyping was performed by PCR using primers flanking deletion breakpoints (Supplementary Table 3). The breakpoints for all F1 pups of stable lines and all F0 embryos from the embryonic screen were sequenced. For the embryonic screen, primers internal to each deletion were used to identify any mosaic embryos carrying both deletion and wild type alleles. For some very small deletions, surveyor assays were used in addition to PCR to exclude mosaicism. The balancer mouse strains DEL(P-F8) and Fgf8[null] were genotyped as previously described[6]. All mouse lines were maintained on a C57Bl/6 background.

**Targeted genome engineering, in vitro fertilization and embryo transfers**. Two CRISPR gRNA targets flanking each region of interest were designed using the CRISPR Design Tool (Zhang Lab, MIT) and are listed in Supplementary Table 4. In vitro transcription and cytoplasmic injections were performed essentially as described previously[61]. Cas9 from px330 (Addgene) was subcloned downstream the T7 promoter in a pGEMte plasmid. The target plasmid was linearized, gel purified and used as template for IVT. Templates for gRNAs were generated through PCR amplification. IVT was performed with mMESSAGE mMACHINE T7 ULTRA kit (Life Technologies) and MEGAshortscript T7 kit (Life Technologies), respectively, and RNA was purified using MEGAclear kit (Life Technologies). Cas9 mRNA (100 ng/μl) and chimeric gRNAs (50 ng/μl) were diluted in micro-injection buffer[62] and injected according to standard procedure. For deletion screening of embryos, in vitro fertilisation (IVF) was performed the night before injections. One DEL(P-F8) heterozygous male was euthanized, the epididymis was dissected out and incubated 25–45 min in fertiup medium at 37 °C, 5% $CO_2$, allowing sperm to swim out. Meanwhile, oocytes from superovulated females were isolated into 200 μl CARD media and 10–20 μl sperm was added before incubation overnight.

**Cloning, transgenesis and X-gal staining**. Transgenesis was performed as previously described[6]. Briefly, fragments of interest were cloned upstream a ß-globin-derived minimal promoter and a LacZ reporter gene. The Tg(DEL-B) and Tg(DEL-C) fragments were cloned from CRISPR-embryo DEL-AB-2 and DEL-C, respectively. Primers used for cloning are listed in Supplementary Table 5. Linearized and gel-purified fragments were microinjected into fertilized mouse oocytes (C57Bl6/J and FVB strain background) and transferred to pseudo-pregnant females (Institute Pasteur, Mouse Genetics Engineering). Embryos were collected at e10.5 and stained for ß-galactosidase activity using standard protocol. Genotyping PCR was performed on yolk sac DNA.

**Optical projection tomography**. Embryonic brains were dissected free at e18.5, fixed in 4% PFA O/N and prepared for OPT scanning[63]. Each specimen was scanned using the Bioptonics 3001 OPT scanner with a resolution of $1024 \times 1024$ pixels and reconstructed with the NRecon version 1.6.9.18 (Skyscan) software. Post-acquisition alignment values for reconstructions were calculated using LLS-Gradient based A-value tuning[64]. Screenshots were exported from OPT volume renderings generated in Drishti v2.6.3[65] and processed in Photoshop CS5 version 9.0.2 (Adobe). All image adjustments were applied equally to entire images and occasional artefacts such as fibres or dust were digitally removed.

**Gene expression analysis**. In situ hybridisation was performed according to standard protocols with previously published Fgf8 probe[66] ($n > 3$ for all experimental groups, with the exception of DEL66 where $n = 3$). For RT-qPCR, the MHB-region was dissected from e10.5 embryos and total RNA was extracted using the RNAeasy (Qiagene) kit. cDNA was prepared using the ProtoScript First Strand cDNA Synthesis Kit (New England Biolabs) with random primers. For each reaction, 150–200 ng RNA was used. RT-PCR was performed according to manufacturers protocol on a GE48.48 IFC (Fluidigm) using SsoFast EvaGreen Super-mix with low ROX (Fluidigm). Before RT-PCR, 10 (MHB) or 14 (limb) cycles of preamplification (Fluidigm PreAmp Master Mix) was performed using 15 ng of input cDNA. Preamplified DNA was diluted 5 (MHB) or 10 (limb) times before RT-PCR reaction. Primers used are listed in (Supplementary Table 6). Statistical analysis was performed using the Prism 8 software.

**Motif analysis**. For phylogenetic footprints, sequences of interest were retrieved from pre-calculated alignments at UCSC or Ensembl genome browsers; realigned using MUSCLE, and PWMs were calculated from these alignments. Motif analysis was performed using the online interface of the MEME suite[67].

**Reporting summary**. Further information on research design is available in the Nature Research Reporting Summary linked to this article.

## Data availability

All relevant data supporting the key findings of this study are available within the Article and its Supplementary Information files or from the corresponding author upon reasonable request. A reporting summary for this Article is available as a Supplementary Information file. Source data are provided with this paper.

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

## Acknowledgements

We acknowledge EMBL gene core facility as well as the Technology Core of the Center for Translational Science (CRT) at Institut Pasteur for support in conducting this study. We thank the DKFZ light microscopy facility and F. Bestvater for access to optical projection tomography equipment. We thank I. Braasch for providing spotted gar DNA. We acknowledge Y. Petersen at EMBL transgene facility as well as the animal facilities at both EMBL Heidelberg and Institut Pasteur. A.H. was supported by an EMBL Inter-disciplinary Postdoctoral fellowship (EIPOD) under Marie Curie Actions COFUND from the E.U. This work was supported by a grant from the Deutsche For-schungsgemeinschaft (SP 1331/3-1, to F.S) and by the Institut Pasteur.

## Author contributions

F.S conceived the project and A.H. and F.S. designed the experimental strategies; A.H. performed or supervised all experiments. K.L. and S.B contributed to mouse embryos injections and transfers, and in situ hybridisation and skeletal preparations, respectively. F.L. produced DEL-B and DEL-C mutant mouse lines as well as transgenic LacZ reporter embryos; A.H. and F.S. wrote the paper with input of all authors.

## Competing interests

The authors declare no competing interests.
