## [Peer Review File · Nature Communications]

Reviewers' comments:

Reviewer #1 (Remarks to the Author):

In the present manuscript, Hörnblad et al. study the regulation of *Fgf8*, an essential factor involved in limb and brain development. By using state-of-the-art transgenic technologies, the authors aim to characterize the *in vivo* functionality of putative enhancers that were identified in a previous study. The results delineate a complex genomic landscape controlling *Fgf8* expression in different tissues, with regulatory elements showing extensive redundancy, and provide clues of how such regulation has been carved through evolution.

In addition, the manuscript highlights a problem often faced when gene regulation is investigated *in vivo*. In such studies, reporter assays are considered as a gold standard test for enhancer characterization as they provide precise spatio-temporal information on regulatory activity. However, those assays do not take genomic context into account therefore ignoring different aspects of gene regulation such as 3D folding, nuclear positioning or DNA methylation. Those aspects can modulate enhancer activity and change dramatically the functionality of such elements. In this sense, functional studies like the presented here are necessary and crucial to understand gene regulation at its best.

The manuscript is well written, easy to follow and the results are clearly presented. This work is in principle adequate for Nature Communications provided that the authors address the following criticisms:

Major comments:

1- In figure 1D, *in situ* hybridization for *Fgf8* is presented for all individual deletion of putative AER enhancers. Despite claiming that there are no changes in expression, Del66 clearly shows spatial as well as quantitative differences. Unfortunately, qPCRs have not been performed for this element. The authors should perform those experiments to rule out the possibility that the deletion of this element has transcriptional consequences.

2- The deletion of the brain enhancer CE68 has an effect on *Fgf8* expression, a gene with a clear role in brain development, but also alters the intron of *Fbxw4*. This gene is also expressed in brain, at least at E14.5 (Diez-Roux et al, PLOS, 2011), and there is not available knockout to my knowledge that can rule out a possible involvement of *Fbxw4* in the observed phenotypes. The authors should, at least, show that the manipulation of *Fbxw4* intron does not affect its expression in brain and argument why they think this gene is not involved in the pathogenic phenotype.

3- There are recent publications that studied the phenomenon of enhancer redundancy *in vivo* also in mammalian systems (Hay et al., Nat Genet, 2016; Shin et al., Nat Genet, 2016; Will et al., Nat Genet, 2017). Those publications should be cited and the results from the current manuscript discussed in the context of that previous work.

Minor comments

1- Line 171 and 177-178. The qPCR data for most of the genes testes is missing from the corresponding figures.

2- Line 299. It is not shown where the PWMs are found in spotted gar and zebrafish CE64 enhancers.

3- Line 395. Table S2 refers to table S4

Reviewer #2 (Remarks to the Author):

In the article 'Dissection of the *Fgf8* regulatory landscape by in vivo CRISPR-editing reveals extensive inter- and intra-enhancer redundancy' the authors systematically delete previously identified regulatory regions around *fgf8* locus, to understand the regulatory landscape of this locus. They show that for the limb there is functional redundancy while for the midbrain-hindbrain boundary (MHB), there are two redundant enhancers and one enhancer, CE64, that is vital for enhancer activity and its removal leads to a MHB phenotype. They then use both CRISPR mutation analyses and evolutionary conservation to further characterize the functional units of this enhancer. The manuscript uses elegant mouse genetics to very nicely show that enhancers can be redundant and non-redundant and follows up with deeper functional characterization of one of these enhancers and as such is a great fit for the broad audience of Nature Communications. It is also very nicely written and easy to follow. I have the following comments:

-The authors demonstrated that deletion of CE64 caused complete loss of midbrain and cerebellum, while CE79 and 80 were not essential, claiming that *Fgf8* expression in the MHB is regulated mainly by CE64. However, they did not pay attention to temporal specificity of these enhancers. Beermann et al (PMID: 16397882) demonstrated that CE79 (CR3 region in their paper) showed enhancer activity at MHB only after E9.0, while CE64 seems to function after E8.25 (based on Fig. 3G-H). As the CE64 seems to be the only enhancer that functions at E8.25-E9.0, the reason why only CE64 was essential for the gene expression in the MHB could be because of its temporal specificity, rather than functional importance among these three enhancers. If that case, their claim, i.e. CE64 is the main enhancer and CE79-80 are accessory enhancers, might be misleading. To see temporal difference of their activities, the authors can show lacZ reporter assays for CE64, 79, and 80 at both E8.25-E8.5 and E9.0-9.5. This will also help understand temporal change of the regulatory network of MHB genes, including *Fgf8*, *Wnt*, *Pax2* (known to bind to CE79 (PMID: 18280464)), etc. It might also be good to quantify *Fgf8* expression via qPCR and in situ at those time points for the various deletions of these enhancers.

-Initially, out of 7 putative enhancers in *fgf8* locus they focus their attention to CE64 for rest of the manuscript. They tested hemizygous enhancer deletions over *Fgf8* null background for phenotypic analysis. It would be good if the authors could comment if the animals carrying deletions that did not show any developmental phenotypes were normal and fertile adults without any other overt phenotype in adulthood.

-The authors found DEL64 to be the major contributing regulatory element for *Fgf8* expression in brain. For their third result section, they did a lot of *fgf8* downstream gene expression analysis in DEL64, 79, 80, 79-80 (I assume in Del hemizygous over *fgf8* null background). I would also like to see *Fbxw8* and neighboring gene expression, for DEL64 condition. The landscape of *Fgf8* locus is gene rich (*Npm3*, *Mgea5*, *Dpcd*, *Pol1*, *Btrc*). To understand its regulation it is crucial to understand how these deletions are affecting other genes in the locus, since many of these genes have similar cell type expression.

- 'Instead of an expected 50% reduction, we measured that *Fgf8* expression in *Fgf8* null/+ was 79% of wild-type level in the MHB, and 68% in the limb' It would be good to mention in the main text the genetic background of these mice due to this and whether this could influence any of the results in general.

-The authors then generated many deletion alleles for DEL64 to 'bash CE64'. In Fig4, they present panel of 14 deletion alleles and one 10kb insertion allele *Dac2J* (to be reported in future manuscript). Out of 2Kb CE64, around 1.2kb of its centromeric end is dispensable, making fragments B+C (around 700bp) to be functional parts of CE64 enhancer, that have high conservation too. Both B+C are absolutely necessary for midbrain formation as seen in *DEIB1* and *DelC1*. There are three functionally redundant blocks within B. This is written well in text but can be depicted better in the figure. The authors can also name these sub-blocks to aid ease of

explanations for their final result section. It would also be good to have text above the last column in Figure 4d that says what X and checkmark means.

-In their last result, the authors tested functional conservation of CE64 by doing reporter assays of corresponding region from fish, spotted gar. The final section where they talk about putative TFBS isn't informative and functional enough to claim, that CE64 use different logics to rewire fgf8 regulatory circuitry in fishes and mammals. It is noteworthy that in zebrafish too the organization of fgf8 and Fbxw4 gene is conserved too. This again calls for analyses of the effect of these deletions on Fbxw4 gene expression and to say that fgf8 regulatory circuitry is tightly conserved. I would tone down these claims.

Reviewer #3 (Remarks to the Author):

This paper is a follow-up to this lab's previous study on enhancers that control Fgf8 expression in the mouse embryo (Dev Cell 24: 530). The authors cogently frame the problem that function (determined by deletions) and activity (determined by transgenics) are not the same. The authors use CRISPR-mediated modifications and transgenesis to explore the AER and MHB enhancers. However, it is mostly a study of the MHB enhancers, with the more superficial AER data added on. Although the AER data could be removed without detracting from the data, it doesn't hurt to include it. If the authors had also provided experimental data for the trans-acting factor(s) driving expression at CE64 (or CE64-C) it would have been a more complete. Nevertheless, as it stands, we found it a very compelling and interesting paper.

Minor points:

- 1) Reference 20: Moon, A. M. and Capecchi, M. R. are listed twice.
- 2) Caption fig1: "The limb AER elicits extensive regulatory redundancy." "Elicits" is a strange word choice, implying active agency to the AER.
- 3) Page 5, line 108: "Previous experiments had demonstrated that mice carrying a deletion of the region containing the four distal enhancers abolishes limb Fgf8 expression and causes similar defects to the conditional ablation of Fgf8 in the limb. " needs a reference. We assume the authors are referring to Figure 3 in Dev Cell 24: 530).
- 4) Page 7, line 186 "that it is required" – needs a grammar correction or typo fix.
- 5) Pg 8, line 221: "Of note, a 10kb insertion of a MusD retro element as observed in the Dac2J strain (Fig4D), appear to have no impact on MHB development (Tugce Aktas, in preparation)." We assume the authors are referring to the vertical dash in the bottom Figure 4D. This dash needs some description in the caption.
- 6) Page 10, line 275 "It is also noteworthy that the zebrafish dr10 enhancer recapitulates the broad activity of mouse CE64 in the MHB region (as well as in the forebrain and tail bud), in the zebrafish transcriptional context" needs a reference.
- 7) Top of page 9, the authors sometimes refer to "Tg(64-DEL-B)" and sometimes "Tg(DEL-B)" in referring to the same transgene. They should be consistent.

Major points:

1) Page 5 line 125: "Taken together, this demonstrates that the regulatory system that controls Fgf8 limb expression in vivo is highly modular and displays extensive regulatory redundancy." This idea is stated too definitively. It would be less speculative if the authors had analyzed combinations of deletions that are shown in Figure 1. The large deletions in Dev Cell 24: 530 are too large to be specific. For example, perhaps only two specific enhancers are required. Or perhaps another region within deletions (from Dev Cell 24: 530), other than one of these enhancers, is required. Therefore.. the authors should change the text to "Taken together, this suggest that the...."

2) Page 6 line 143: "Despite the sensitivity of the MHB-derived structures to mild-reduction of Fgf8-signalling from the IsO, which could result in various degrees of hypoplasia 17,28, the two proximal enhancers 79 and 80 appear dispensable for the development of those structures." This conclusion comes at an odd point in the narrative because all we know is that deletion of enhancers 79 and 80 (over the null) have no MHB defect. However, once the reader knows that these genotypes have a diminution in Fgf8 expression (in Figure 3i), then this idea makes sense. So, it should be moved forward.

3) In the AER section the author state that they characterized their CRISPR-mediated deletions over an Fgf8 null allele. When they first describe the phenotype of their MHB deletions (and refer to Figure 2), the results text is not clear whether these deletions are also over the null or are homozygous. Please correct this.

4) Regarding the mhb enhancer deletions: are DEL79 and DEL80 animals viable? Do DEL64 embryos show aberrant cell death, as described for a MHB-specific deletion of Fgf8 (Development 130, 2633)?

5) Bravo to the authors for looking at Fgf8 mRNA levels in null heterozygotes. It is interesting that they found levels above 50%. However, their explanation for the MHB data- that negative feedback components (Spry2 and Dusp6) are reduced doesn't make sense because Fgf8 is not a downstream target of the FGF8 signaling pathway in the MHB – see Figure 5i, J in (Development 130, 2633).

6) In Figure 3G,H , Fgf8 appears to be somewhat reduced in the forebrain and primitive streak. Is this reproducible? If so the author should comment on it. If not, they should find another sample for the figure.

7) The data in Figure 5b needs more samples for statistical significance. Using Fisher's exact test, Tg(DEL-B) is significantly different from Tg(CE64) (0/8 vs 3/4) but the other two constructs are not. Assuming the numerator remains zero, the authors need to look at least 5 samples.

8) "Noteworthy, the degree of conservation of the different AER enhancer does not seem to correlate with relative importance," We don't understand how the authors can say this. There is no data in this paper regarding the degree of conversation nor any experiments that determine the relative importance of the enhancers. Unless we are missing something, this sentence should be removed.

Responses to Reviewers' comments:

Reviewer #1 (Remarks to the Author):

In the present manuscript, Hörnblad et al. study the regulation of *Fgf8*, an essential factor involved in limb and brain development. By using state-of-the-art transgenic technologies, the authors aim to characterize the *in vivo* functionality of putative enhancers that were identified in a previous study. The results delineate a complex genomic landscape controlling *Fgf8* expression in different tissues, with regulatory elements showing extensive redundancy, and provide clues of how such regulation has been carved through evolution.

In addition, the manuscript highlights a problem often faced when gene regulation is investigated *in vivo*. In such studies, reporter assays are considered as a gold standard test for enhancer characterization as they provide precise spatio-temporal information on regulatory activity. However, those assays do not take genomic context into account therefore ignoring different aspects of gene regulation such as 3D folding, nuclear positioning or DNA methylation. Those aspects can modulate enhancer activity and change dramatically the functionality of such elements. In this sense, functional studies like the presented here are necessary and crucial to understand gene regulation at its best.

The manuscript is well written, easy to follow and the results are clearly presented. This work is in principle adequate for Nature Communications provided that the authors address the following criticisms:

Major comments:

1- In figure 1D, *in situ* hybridization for *Fgf8* is presented for all individual deletion of putative AER enhancers. Despite claiming that there are no changes in expression, Del66 clearly shows spatial as well as quantitative differences. Unfortunately, qPCRs have not been performed for this element. The authors should perform those experiments to rule out the possibility that the deletion of this element has transcriptional consequences.

The reviewer is correct that the photo displayed for DEL66 in Fig1D may give the wrong impression (particularly in the hindlimb) that *Fgf8* expression is reduced. This is however not representative of all different embryos and may have resulted from the angle or slightly earlier stage. To correct for this, we have now added next to each embryo a view of the limbs of the corresponding littermate, which shows that expression patterns are not different between WT and 66DEL embryos at the same stage.

2- The deletion of the brain enhancer CE68 has an effect on *Fgf8* expression, a gene with a clear role in brain development, but also alters the intron of *Fbxw4*. This gene is also expressed in brain, at least at E14.5 (Diez-Roux et al, PLOS, 2011), and there is not available knockout to my knowledge that can rule out a possible involvement of *Fbxw4* in the observed phenotypes. The authors should, at least, show that the manipulation of *Fbxw4* intron does not affect its

expression in brain and argument why they think this gene is not involved in the pathogenic phenotype.

As pointed out by Reviewer #1, a valid question is whether any of the effects seen in our mutants can be due to effects on *Fbxw4* expression or function. The complete lack of MHB in the DEL64 mutant precludes the analysis of quantitative gene expression changes in these cells. However, there are several lines of evidence indicating that *Fbxw4* is not involved in the pathogenic phenotype.

First, Marinic et al. 2013 characterized the gene expression changes resulting from a deletion spanning a large part of the *Fbxw4* gene including the enhancers CE58-CE66. The phenotypes described therein, which are all similar to described *Fgf8* KO phenotypes, were identical regardless of whether the deletion was analyzed over a *Fgf8*_{null} allele that retained an intact *Fbxw4* copy or a large deletion allele that removed both *Fbxw4* and *Fgf8*. Furthermore, deletion of CE64 or the entire enhancer cluster CE58-CE66 do not display any abnormalities in heterozygous condition. Thus, the phenotypes seen are not likely to result from lack of *Fbxw4* expression.

In addition, DEL61 and DEL66 as well as the previously described Dac2J strain, containing a 10kb insertion in CE64, all contain mutations manipulating the same *Fbxw4* intron as CE64 and none display abnormal development of the MHB. This is also true for the embryonic deletion of 64-A. Taken together, these data argue that *Fbxw4* is not involved in the pathogenic phenotype. This is now mentioned in the discussion section of the manuscript.

3- There are recent publications that studied the phenomenon of enhancer redundancy in vivo also in mammalian systems (Hay et al., Nat Genet, 2016; Shin et al., Nat Genet, 2016; Will et al., Nat Genet, 2017). Those publications should be cited and the results from the current manuscript discussed in the context of that previous work.

The results are now discussed in the context of these publications as well as the recent paper from Pennacchio lab (Osterwalder et al, Nature, 2018). The reference list has been updated accordingly.

Minor comments

1- Line 171 and 177-178. The qPCR data for most of the genes testes is missing from the corresponding figures.

This qPCR data has now been included.

2- Line 299. It is not shown where the PWMs are found in spotted gar and zebrafish CE64 enhancers.

Supplementary FigS8 includes now the localization of PWMs in spotted gar and zebrafish CE64 enhancers.

3- Line 395. Table S2 refers to table S4

This typo has now been corrected.

Reviewer #2 (Remarks to the Author):

In the article 'Dissection of the Fgf8 regulatory landscape by in vivo CRISPR-editing reveals extensive inter- and intra-enhancer redundancy' the authors systematically delete previously identified regulatory regions around *fgf8* locus, to understand the regulatory landscape of this locus. They show that for the limb there is functional redundancy while for the midbrain-hindbrain boundary (MHB), there are two redundant enhancers and one enhancer, CE64, that is vital for enhancer activity and its removal leads to a MHB phenotype. They then use both CRISPR mutation analyses and evolutionary conservation to further characterize the functional units of this enhancer. The manuscript uses elegant mouse genetics to very nicely show that enhancers can be redundant and non-redundant and follows up with deeper functional characterization of one of these enhancers and as such is a great fit for the broad audience of Nature Communications. It is also very nicely written and easy to follow. I have the following comments:

-The authors demonstrated that deletion of CE64 caused complete loss of midbrain and cerebellum, while CE79 and 80 were not essential, claiming that *Fgf8* expression in the MHB is regulated mainly by CE64. However, they did not pay attention to temporal specificity of these enhancers. Beermann et al (PMID: 16397882) demonstrated that CE79 (CR3 region in their paper) showed enhancer activity at MHB only after E9.0, while CE64 seems to function after E8.25 (based on Fig. 3G-H). As the CE64 seems to be the only enhancer that functions at E8.25-E9.0, the reason why only CE64 was essential for the gene expression in the MHB could be because of its temporal specificity, rather than functional importance among these three enhancers. If that case, their claim, i.e. CE64 is the main enhancer and CE79-80 are accessory enhancers, might be misleading. To see temporal difference of their activities, the authors can show lacZ reporter assays for CE64, 79, and 80 at both E8.25-E8.5 and E9.0-9.5.

This will also help understand temporal change of the regulatory network of MHB genes, including *Fgf8*, *Wnt*, *Pax2* (known to bind to CE79 (PMID: 18280464)), etc. It might also be good to quantify *Fgf8* expression via qPCR and in situ at those time points for the various deletions of these enhancers.

We agree with the reviewer that this is a very relevant point, which we now have addressed through transient transgenesis of the three different enhancers. These results have been added to the manuscript in the section "Temporal specificity of CE64 underlies initiation of *Fgf8* gene expression". The results indeed show that the activity of CE64 precedes the other two MHB enhancers, something that could explain the DEL64 phenotype and its importance at this stage. Neither CE79 nor CE80 display enhancer activity in the early phase of *Fgf8* expression, and therefore deletions of these elements are neither likely to affect gene expression at this stage.

Our analyses of mice lacking CE79, CE80 or both together, clearly show that these enhancers are dispensable for the specification of the MHB and

subsequent formation of the midbrain and anterior hindbrain. We cannot formally rule out that CE79 and/or CE80 would be sufficient for the maintenance of *Fgf8* expression after initial induction, but none of these enhancer mutants manifest changes in *Fgf8* gene expression profiles that would indicate such a scenario. Thus, we believe it is fair to say that CE64 is the main enhancer in this system, and that part of its functional relevance comes from its temporal specificity. However, we have rephrased the wording to clarify that “CE64 is required and sufficient for proper initiation of *Fgf8* expression and sufficient for subsequent maintenance” to tone down the claims of its importance at later stages.

-Initially, out of 7 putative enhancers in *fgf8* locus they focus their attention to CE64 for rest of the manuscript. They tested hemizygous enhancer deletions over *Fgf8*^{null} background for phenotypic analysis. It would be good if the authors could comment if the animals carrying deletions that did not show any developmental phenotypes were normal and fertile adults without any other overt phenotype in adulthood.

All lines carrying putative enhancer deletions, with the exception of the CE64 deletion were healthy and fertile in homozygous condition. Adult *Fgf8*^{null/DEL79}, *Fgf8*^{null/DEL80}, and *Fgf8*^{null/DEL79-80} were also healthy, without any apparent phenotype. This has now been clarified in the manuscript.

-The authors found DEL64 to be the major contributing regulatory element for *Fgf8* expression in brain. For their third result section, they did a lot of *fgf8* downstream gene expression analysis in DEL64, 79, 80, 79-80 (I assume in Del hemizygous over *fgf8* null background). I would also like to see *Fbxw8* and neighboring gene expression, for DEL64 condition. The landscape of *Fgf8* locus is gene rich (*Npm3*, *Mgea5*, *Dpcd*, *Pol1*, *Btrc*). To understand its regulation it is crucial to understand how these deletions are affecting other genes in the locus, since many of these genes have similar cell type expression.

This is an important point that was also raised by reviewer #1 - See comment above (Point 2). Unfortunately, analyzing gene expression levels in the DEL64 mutant is not possible as the MHB cells we are interested in are never specified and hence not present.

- ‘Instead of an expected 50% reduction, we measured that *Fgf8* expression in *Fgf8*^{null/+} was 79% of wild-type level in the MHB, and 68% in the limb’ It would be good to mention in the main text the genetic background of these mice due to this and whether this could influence any of the results in general.

A paragraph has been added in the discussion to keep in mind that differences in genetic background may very well influence the importance of individual enhancers on gene expression. In the methods section, under the “Animals and genotyping” subheading, we have now mentioned that the genetic background of the strain used is C57Bl/6.

-The authors then generated many deletion alleles for DEL64 to ‘bash CE64’. In Fig4, they present panel of 14 deletion alleles and one 10kb insertion allele

Dac2J (to be reported in future manuscript). Out of 2Kb CE64, around 1.2kb of its centromeric end is dispensable, making fragments B+C (around 700bp) to be functional parts of CE64 enhancer, that have high conservation too. Both B+C are absolutely necessary for midbrain formation as seen in DEIB1 and DelC1. There are three functionally redundant blocks within B. This is written well in text but can be depicted better in the figure. The authors can also name these sub-blocks to aid ease of explanations for their final result section. It would also be good to have text above the last column in Figure 4d that says what X and checkmark means.

We agree with the reviewer that this could be better depicted in the figure. The sub-blocks have been named R1-R3 in Fig4E and Fig7B and are referred to as such in the text. The caption for tickmark and X has also been moved from 4B to 4D to clarify their meaning.

-In their last result, the authors tested functional conservation of CE64 by doing reporter assays of corresponding region from fish, spotted gar. The final section where they talk about putative TFBS isn't informative and functional enough to claim, that CE64 use different logics to rewire fgf8 regulatory circuitry in fishes and mammals. It is noteworthy that in zebrafish too the organization of fgf8 and Fbxw4 gene is conserved too. This again calls for analyses of the effect of these deletions on Fbxw4 gene expression and to say that fgf8 regulatory circuitry is tightly conserved. I would tone down these claims.

The finding that the core conserved sequence from spotted gar can drive expression in the mouse MHB in transgenic assays as opposed to its mouse counterpart is intriguing and suggests that there are qualitative differences between these conserved sequences from the two species. Still, as reviewer #2 points out, this does not necessarily mean that a rewiring of the regulatory network has occurred, and the conservation of the locus in conjunction with the similar reporter activities of the whole enhancer in respectively species suggests that the overall function of this enhancer in the MHB is conserved. Therefore, these claims have been toned down and the wording in the last paragraph has been changed accordingly.

Reviewer #3 (Remarks to the Author):

This paper is a follow-up to this lab's previous study on enhancers that control Fgf8 expression in the mouse embryo (Dev Cell 24: 530). The authors cogently frame the problem that function (determined by deletions) and activity (determined by transgenics) are not the same. The authors use CRISPR-mediated modifications and transgenesis to explore the AER and MHB enhancers. However, it is mostly a study of the MHB enhancers, with the more superficial AER data added on. Although the AER data could be removed without detracting from the data, it doesn't hurt to include it. If the authors had also provided experimental data for the trans-acting factor(s) driving expression at CE64 (or CE64-C) it would have been a more complete. Nevertheless, as it stands, we found it a very compelling and interesting paper.

Minor points:

1) Reference 20: Moon, A. M. and Capecchi, M. R. are listed twice.

This has been corrected.

2) Caption fig1: “The limb AER elicits extensive regulatory redundancy.” “Elicits” is a strange word choice, implying active agency to the AER.

The wording has been changed to “presents”.

3) Page 5, line 108: “Previous experiments had demonstrated that mice carrying a deletion of the region containing the four distal enhancers abolishes limb Fgf8 expression and causes similar defects to the conditional ablation of Fgf8 in the limb. “ needs a reference. We assume the authors are referring to Figure 3 in Dev Cell 24: 530).

This reference has been now added (Marinic et al. 2013).

4) Page 7, line 186 “that it is required” – needs a grammar correction or typo fix.

The last part of this paragraph has been rephrased to clarify the possible roles of the different enhancers and this sentence has been removed.

5) Pg 8, line 221: “Of note, a 10kb insertion of a MusD retro element as observed in the Dac2J strain (Fig4D), appear to have no impact on MHB development (Tugce Aktas, in preparation).” We assume the authors are referring to the vertical dash in the bottom Figure 4D. This dash needs some description in the caption.

The description for this caption has now been added in Figure 4D.

6) Page 10, line 275 “It is also noteworthy that the zebrafish dr10 enhancer recapitulates the broad activity of mouse CE64 in the MHB region (as well as in the forebrain and tail bud), in the zebrafish transcriptional context” needs a reference.

The reference Komisarczuk et al 2009 has now been added to this sentence.

7) Top of page 9, the authors sometimes refer to “Tg(64-DEL-B)” and sometimes “Tg(DEL-B)” in referring to the same transgene. They should be consistent.

This has been corrected now for consistency.

Major points:

1) Page 5 line 125: “Taken together, this demonstrates that the regulatory system that controls Fgf8 limb expression in vivo is highly modular and displays

extensive regulatory redundancy.” This idea is stated too definitively. It would be less speculative if the authors had analyzed combinations of deletions that are shown in Figure 1. The large deletions in Dev Cell 24: 530 are too large to be specific. For example, perhaps only two specific enhancers are required. Or perhaps another region within deletions (from Dev Cell 24: 530), other than one of these enhancers, is required. Therefore.. the authors should change the text to “Taken together, this suggest that the....”

We have now toned down this claim and changed the wording to “suggest”

2) Page 6 line 143: “Despite the sensitivity of the MHB-derived structures to mild-reduction of Fgf8-signalling from the IsO, which could result in various degrees of hypoplasia 17,28, the two proximal enhancers 79 and 80 appear dispensable for the development of those structures.” This conclusion comes at an odd point in the narrative because all we know is that deletion of enhancers 79 and 80 (over the null) have no MHB defect. However, once the reader knows that these genotypes have a diminution in Fgf8 expression (in Figure 3i), then this idea makes sense. So, it should be moved forward.

This is a good suggestion from reviewer #3 and this sentence have now been moved forward.

3) In the AER section the author state that they characterized their CRISPR-mediated deletions over an Fgf8 null allele. When they first describe the phenotype of their MHB deletions (and refer to Figure 2), the results text is not clear whether these deletions are also over the null or are homozygous. Please correct this.

This has now been clarified in the manuscript in the first paragraph of the results section.

4) Regarding the mhb enhancer deletions: are DEL79 and DEL80 animals viable? Do DEL64 embryos show aberrant cell death, as described for a MHB-specific deletion of Fgf8 (Development 130, 2633)?

Yes, all lines (mhb and limb) except DEL64 are viable and fertile in homozygosis. This is now clarified in the manuscript. In the DEL64 mutants we have not assessed cell death in the MHB region as it is already well documented that loss or moderate reduction of Fgf8 expression leads to subsequent cell death in the region (Basson 2008, Chi 2003, Guo 2010).

5) Bravo to the authors for looking at Fgf8 mRNA levels in null heterozygotes. It is interesting that they found levels above 50%. However, their explanation for the MHB data- that negative feedback components (Spry2 and Dusp6) are reduced doesn't make sense because Fgf8 is not a downstream target of the FGF8 signaling pathway in the MHB – see Figure 5i, J in (Development 130, 2633).

It is true that this explanation is merely speculative so we have toned down the wording of this point. Still, both Dusp6 and Spry2 are expressed in the MHB

region and are described as negative feedback modulators of Fgf8 signaling. Overexpression of Spry2 in the region leads to downregulation of Fgf8 (Basson et al 2008). Dusp6 is also expressed in the region and its expression is reduced when Fgf-signalling is perturbed. As reviewer #3 points out, Chi et al 2003 shows that transcription of the truncated *Fgf8* allele still occurs at 7 somites in their conditional mutant. They therefore conclude that *Fgf8* is not autoregulated. However, these mice still have residual transcripts of normal *Fgf8* mRNA at 7s. At 16 somites, an earlier stage than analyzed in our study, transcription of the truncated *Fgf8* allele is not detectable. Thus, during the maintenance phase of expression, *Fgf8* may be subject to autoregulatory feedback.

6) In Figure 3G,H , Fgf8 appears to be somewhat reduced in the forebrain and primitive streak. Is this reproducible? If so the author should comment on it. If not, they should find another sample for the figure.

It is indeed interesting as CE64 also give a broad expression in the forebrain in transgenic assays. We do not see any obvious forebrain phenotype in these mice. We agree that this should be pointed out and have now added this in the manuscript.

7) The data in Figure 5b needs more samples for statistical significance. Using Fisher's exact test, Tg(DEL-B) is significantly different from Tg(CE64) (0/8 vs $\frac{3}{4}$) but the other two constructs are not. Assuming the numerator remains zero, the authors need to look at least 5 samples.

We understand that from a strictly statistical point of view it would have been good to have additional embryos for Tg(DEL-C) and Tg(64-B). Still, despite the difference in length, both Tg(DEL-C) and Tg(64-B) lack the same functional unit 64-C. That means that 0 out of 7 embryos lacking the C unit display expression in the MHB which do give statistical significance.

Additionally, in the embryonic screen for DEL-B and DEL-C we have 5 and 11 alleles, respectively. We also have 2 founder lines of each of these subunits, in which we have confirmed these results. Both embryonic alleles and lines give the same phenotype with lack of the midbrain and the cerebellum and *Fgf8* gene expression is abolished in the MHB in the DEL-B and DEL-C stable lines. In view of these results altogether, we believe the data should be sufficient to support the claim of the transgenic results.

8) "Noteworthy, the degree of conservation of the different AER enhancer does not seem to correlate with relative importance," We don't understand how the authors can say this. There is no data in this paper regarding the degree of conservation nor any experiments that determine the relative importance of the enhancers. Unless we are missing something, this sentence should be removed.

The sentence has now been removed.

REVIEWERS' COMMENTS

Reviewer #1 (Remarks to the Author):

All my concerns have been properly addressed.

Reviewer #2 (Remarks to the Author):

The authors have nicely addressed all my comments.

Reviewer #3 (Remarks to the Author):

The authors have addressed my concerns.

-Mark Lewandoski

Point-by-point response to the reviewers' comments

Please, see reviewers comments below, where no additional points to address have been raised. We would like to thank the reviewers for the helpful comments and constructive input that has helped to improve the quality of this manuscript during the review process.

Reviewer #1 (Remarks to the Author):

All my concerns have been properly addressed.

Reviewer #2 (Remarks to the Author):

The authors have nicely addressed all my comments.

Reviewer #3 (Remarks to the Author):

The authors have addressed my concerns.
-Mark Lewandoski

We thank the reviewers for their help and suggestions.